# Removal of Organic Pollutants with Polylactic Acid-Based Nanofiber Composites

**DOI:** 10.3390/polym14214622

**Published:** 2022-10-31

**Authors:** Dengbang Jiang, Xiushuang Song, Heng Zhang, Mingwei Yuan

**Affiliations:** Green Preparation Technology of Biobased Materials National &Local Joint Engineering Research Center, Yunnan Minzu University, Kunming 650500, China

**Keywords:** polylactic acid, nanofiber, nano-titanium dioxide, graphene oxide, organic pollutants

## Abstract

In the process of using nano-titanium dioxide (TiO_2_) photocatalytic treatment of organic polluted liquid, the easy aggregation and recycling difficulty of nano-TiO_2_ particles are important problems that cannot be avoided. Anchoring nano-TiO_2_ to the substrate not only limits the aggregation of nano-TiO_2_, but also facilitates the easy removal and reuse of nano-TiO_2_ after processing. Herein, coaxial electrospun nanofibrous (NFs) made of L-polylactic acid (PLLA) and chitosan (CS) are coated with graphene oxide (GO) and TiO_2_ for the enhanced oxidation of organic pollutants. The adsorption and photocatalysis experiment results show that, for methyl orange (MO) dye solution, the saturated removal of MO by PLLA/CS, PLLA/CS-GO and PLLA/CS-GO/TiO_2_ nanofibers are 60.09 mg/g, 78.25 mg/g and 153.22 mg/g, respectively; for the Congo red (CR) dye solution, the saturated removal of CR by PLLA/CS, PLLA/CS-GO and PLLA/CS-GO/TiO_2_ nanofiber materials were 138.01 mg/g, 150.22 mg/g and 795.44 mg/g, respectively. These three composite nanofiber membrane materials can maintain more than 80% of their adsorption capacity after four repeated cycles. They are environmentally friendly and efficient organic pollution remediation materials with promising application.

## 1. Introduction

Organic dyes that are widely discharged into the aqueous environment have a serious impact on human health, with cases of teratogenicity and cancer diagnoses becoming more commonplace [1,2]. As a result, researchers have become committed to finding a material with high adsorption capacity that can adsorb pollutants from the aqueous environment without having any negative impact [3].

To this aim, León et al. [4] prepared oxidized chitosan (CS) via thermal acid oxidation and redox reactions, which has the ability to adsorb both cationic and anionic dyes. As activated carbon (AC) is a cost-effective material for treating organic dye solutions in water, Xiao et al. [5] prepared a sugar-based AC material that exhibits high adsorption of rhodamine B (RhB). Sattar et al. [6] prepared polylactic acid (PLA)/AC that shows high adsorption of rhodamine B (RhB). The use of functional nanocomposites (such as carbon nanotubes, graphene, mesoporous materials, etc.) to remove organic pollutants has proved to be an effective means. Niculescu et al. have done a lot of excellent frontier research work in this field [7]. Moreover, graphene oxide (GO) has become increasingly important in the adsorption of organic dyes [8,9]. Omidi et al. [10] prepared konjac glucan/GO hydrogels for the adsorption of methyl orange (MO) and methyl blue, using calcium oxide as a cross-linking agent, where the addition of GO was found to substantially improve the adsorption capacity of the adsorbent towards the dyes. The graphene used in the treatment of dye-polluted wastewater is mostly in the form of particles [11] that are difficult to recover, meaning that it is present in bodies of water for long time periods, thus increasing the burden of purification of the body of water. Furthermore, the graphene particles used in the treatment agglomerate lead to poor adsorption of pollutants from the body of water [12]. Moreover, carriers that can be loaded with graphene are mainly polyethylene- and polyphenylene-based polymers that are hard to degrade, thus representing a form of secondary pollution to the environment [13,14,15,16].

Nano-TiO_2_ [17,18], due to its high specific surface area and high reactivity [19,20,21], has multifunctional ability to oxidize and adsorb organic dyes, which has attracted interest in the treatment of inorganic and organic wastewater. To improve the photocatalytic efficiency of TiO_2_, TiO_2_ has been paired with graphitic materials such as graphene oxide (GO) and reduced graphene oxide (rGO) [22,23]. The electrical conductivity of GO helps electrons stay away in ultraviolet (UV) light TiO_2_, thereby enhancing the photocatalytic activity of TiO_2_. In addition, studies have demonstrated that anatase-type TiO_2_ with exposed {001} planes has higher catalytic activity [24,25]. Niculescu et al. [26,27,28,29] used rice husk ash to prepare silica mesoporous materials and used them as catalyst carriers. Experiments showed that this catalyst can efficiently catalyze the oxidation of organic pollutants such as rhodamine.

Electrospinning nanofibers are attractive due to their high specific surface area [30], high porosity [31], tunable diameter [32], simple production [33], and diversity of polymers compatible with electrospinning [34,35,36] becoming a promising scaffold for nanocomposites. A significant advantage of electrospun fibers compared to other substrates is their high specific surface area; when decorated with nanoparticles, the fibers allow the nanoparticles to maintain a high specific surface area [37]. This maximization of specific surface area helps to improve the catalytic and adsorption efficiency of the material while minimizing the economic and environmental costs of nanomaterial production, use, and end-of-life [38].

Chitosan is a derivative of crustacean shells, and is also a waste from fisheries. Since the primary amine groups on its surface can be functionalized with nanomaterials, chitosan is an excellent choice for the preparation of nanoparticle-loaded electrospinning fibers. For example, the primary amine functional group of chitosan is bound to the epoxy group on the surface through a nucleophilic substitution reaction [39]. However, due to the high viscosity and swelling properties of chitosan in solution, it is difficult to perform electrospinning alone. Therefore, it is often paired with other polymers (such as PLGA, and PLA) [40,41,42] to generate stable fibers that are still capable of functionalization through the amine groups of chitosan.

The work in this paper combines the advantages of GO-TiO_2_ nanocomposites and chitosan-based electrospinning fibers to prepare nanocomposite fibers for enhanced oxidation of organic pollutants. Biodegradable PLLA and CS were prepared by coaxial spinning into nanofiber mats with PLLA as the core layer and CS as the shell layer, which were subsequently functionalized with GO-TiO_2_ nanocomposites. The primary amine groups of the fiber shell (CS) were used as anchor sites for the graphene oxide sheets, which were then decorated with TiO_2_. Under UV irradiation, the ability of nanocomposite fibers to catalyze and adsorb MO and CR for multiple cycles was evaluated, avoiding the expensive and energy-intensive process of recycling and regenerating nanomaterials in suspension. Through our unique pairing of nanocomposites, we realize the potential real-world application of GO-TiO_2_ nanocomposites in decentralized organic wastewater treatment systems.

## 2. Experimental

### 2.1. Materials

L-polylactic acid (PLLA, MW: 130,000–140,000 Da) was prepared in the laboratory using the ring-opening polymerization of lactide [43], and graphene oxide (GO) was prepared in the laboratory using Hummers’ method [44]. Chitosan (CS, 98%, WM: 100,000 Da, 90%deacetylated, 200–500 mPa·s 0.5% in acetic acid, TCI, Portland, OR, America), graphene (98%, Sigma-Aldrich, Shanghai, China), tetrabutyl titanate (TBOT, 98%, Sigma-Aldrich, Shanghai, China), hexafluoroisopropanol (HFIP, 99.8%, Titan Reagent Co., Ltd, Shanghai, China), formic acid (FA, 96%, Titan Reagent Co., Ltd, Shanghai, China), hydrofluoric acid (HF, 48–51% solution in water, Titan Reagent Co., Ltd, Shanghai, China), Congo Red (CR, Damao Reagent Co., Ltd, Tianjin, China) and Methyl Orange (MO, Damao Reagent Co., Ltd, Tianjin, China) were purchased from external reagent companies.

### 2.2. Instrumentation

The equipment employed in this study was an electrospinning machine (YFSP-T, Yunfan Technology, Tianjin, China), a scanning electron microscope (SEM, Nova NANOSEM-450, FEI, Oregon, America), a Transmission Electron Microscopy (TEM, Tecnai-GZ-F30S-TWIN, FEI, Eindhoven, Netherlands), a Energy Dispersive X-ray Spectrometer (EDX, S2 PUMA Series II, Bruker, Massachusetts, America), an X-ray powder diffractometer (XRD, D8-ADVANCE-A25X, Bruker, Massachusetts, America), a UV–visible spectrophotometer (UV, 2600i, SHIMADZU, Kyoto, Japan), a transmission electron microscope (TEM, Tecnai GZ-F30S-TWIN, FEI, Eindhoven, Netherlands), an attenuated total reflection Fourier transform infrared spectrometer (ATR-FTIR, Nicolet 6700, Thermo, MA, USA), a Thermogravimetric analyzer (TA, STA449F3, NETZSCH, Bavaria, Germany) and a photochemical reactor (JP-GHX-II, Jiuping, Wuxi, China).

### 2.3. Preparation of Electrospun PLLA/CS Nanofibrous and Their Loading with GO

PLLA (0.3 g) was dissolved in HFIP (4 mL) to prepare a nucleus layer solution with a concentration of 0.075 g mL^−1^ Separately, CS (0.15 g) was dissolved in a mixed solution of HFIP (6 mL) and FA (1 mL) to prepare a shell solution with a concentration of 0.021 g mL^−1^. Then, the prepared solutions were loaded into the core and shell solution syringes (5 mL) corresponding to the coaxial needles (20 G, inner diameter 0.5 mm, outer diameter 0.9 mm), respectively. We set the DC voltage supplier (Gamma High voltage Research Inc.) to a potential of 17.5 kV, connected it to the tip of the needle, and ground it to an aluminum foil-covered roller at a distance of 12 cm. The roller was then rotated at approximately 15 cm min^−1^ to act as a fiber collector, collecting the shell (0.2 mL h^−1^) and core (0.4 mL h^−1^) polymer solutions flowing from the syringe needle, electrospun at room temperature (25 °C) and 40% humidity for 10 h, and then the electrospun nanofibrous (NFs) were rinsed thoroughly with deionized water and dried in a vacuum drying oven (40 °C) for 12 h.

A mixture of GO (4 mg) and deionized (DI) water (15 mL) was treated in an ultrasonic water bath (Controlled ultrasonic cleaning machine, KH-400KDE, Kunshan, China) at 400 W, 40 kHz for 10 min until GO was uniformly and stably dispersed in the water. The NFs (30 mg, 1 mL of GO suspension per 2 mg of fiber) were then fully immersed in the GO suspension in an ultrasonic water bath (400 w, 40 kHz) for 30 min to complete the loading reaction. The loading of GO on NFs was accomplished by epoxyamine addition, in which the epoxy ring on the surface of GO was opened and combined with primary amine groups in chitosan [39]. After loading was completed, the NFs were rinsed with DI water to remove loosely bound GO and dried in a vacuum drying oven (40 °C) for 12 h. The resulting product is denoted as GO@NFs.

### 2.4. Preparation of GO/TiO_2_ and Its Loading on the PLLA/CS NFs

The mixture of GO (8 mg) and absolute ethanol (15 mL) was sonicated (400 w, 40 kHz) for 30 min to obtain a homogeneous suspension. Ti(OBu)_4_ (1 mL) was added to the GO suspension, and the mixture was further sonicated (400 w, 40 kHz) for 30 min to form a homogeneous suspension.

To this suspension was added hydrofluoric acid (HF, 0.3 mL) with continuous stirring, and then the solution was transferred to an autoclave and reacted at 180 °C for 12 h. After the reaction was completed, the mixture was allowed to cool to room temperature, and the produced material was then washed alternately with deionized water and absolute ethanol twice, then centrifuged (6000 r min^−1^, 20 min), filtered, and the resultant powder was placed in a vacuum drying oven after drying at 60 °C for 12 h. The resulting product is denoted as GO/TiO_2_.

A mixture of GO/TiO_2_ (4 mg) and deionized (DI) water (15 mL) was treated in an ultrasonic water bath at 400 w, 40 kHz for 10 min until GO/TiO_2_ was uniformly and stably dispersed in the water. The NFs (15 mg, 1 mL of GO/TiO_2_ suspension per 1 mg of fiber) were then fully immersed in the GO/TiO_2_ suspension in an ultrasonic water bath (400 w, 40 kHz) for 30 min to complete the loading reaction. After loading was completed, the NFs were rinsed with DI water to remove loosely bound GO/TiO_2_ and dried in a vacuum drying oven (40 °C) for 12 h. The resulting product is denoted as GO/TiO_2_@NFs.

### 2.5. Characterization of Fiber Composites

The surface morphology of fibers before and after modification with nanocomposites was analyzed by sputter-coated fibers with 8 nm iridium (Cressington 208HR), and then imaged on SEM at a beam accelerating voltage of 10 kV. The presence of titanium on the nanocomposite fiber surfaces was characterized using TEM and EDX paired with SEM, operating at an accelerating voltage of 200 kV. The distribution of fiber diameters was determined by measuring the cross-section of 100 fibers from the SEM images of the fibers by ImageJ software. The binding mechanism of TiO_2_ to the nanocomposite fibers was characterized by attenuated total reflection Fourier transform infrared spectroscopy from 450 cm^−1^ to 4000 cm^−1^, and the percentage of TiO_2_ loading on the nanocomposite fibers was quantified using thermogravimetric analysis from room temperature to 800 °C.

### 2.6. Adsorption of CR and MO Dyes on PLLA/CS–GO

Portions of dried composite NFs (50 mg) were weighed and added to neutral Congo Red (CR) (0.4 mg mL^−1^, 25 mL) and Methyl Orange (MO) (0.24 mg mL^−1^, 25 mL) dye solutions, respectively, under magnetic stirring. The optimal adsorption temperature and adsorption saturation time of composite fibers for CR and MO dyes were studied by changing the solution temperature and sampling periodically (interval time 0.5 h).

### 2.7. Photodegradation of CR and MO Dyes on PLLA/CS–GO/TiO2

(1) PLLA/CS–GO/TiO_2_ (50 mg) was added to a solution of CR dye (1.6 mg mL^−1^, 25 mL) and the resultant mixture stirred in a dark box for 1 h to establish adsorption–desorption equilibrium prior to being exposed to UV light irradiation. The UV wavelength used in the experiment is 350 nm. An aliquot of the solution was taken up into a quartz tube every 1 h, the absorbance of which was measured by centrifuging the upper layer of the dye solution and using a UV–vis spectrophotometer to study the photodegradation of the CR. (2) NFs (50 mg) were added into CR solutions (0.8–2 mg mL^−1^, 25 mL), and after adsorption–desorption equilibration, the saturation time of adsorption was measured from step (1) by illumination to study the effect that different initial concentrations have on the adsorption performance and the maximum amount of CR adsorbed by the NFs. The dye solutions were then replaced with MO solutions, and the above steps were repeated, with a change in concentrations in step (1) to 0.24 mg mL^−1^, 25 mL of MO and in step (2) to (0.08–0.4 mg mL^−1^), 25 mL of MO.

### 2.8. Cyclic Adsorption Experiments

50 mg PLLA/CS-GO/TiO_2_ was added to 25 mL of neutral CR staining solution with a concentration of 1.2 mg/mL and a neutral MO staining solution with a concentration of 0.32 mg/mL, respectively. After adsorption in the photocatalytic reactor for 3 h, the concentrations of CR and MO in the solution were measured, and the removal rates of CR and MO were calculated. Then the PLLA/CS-GO/TiO_2_ membranes adsorbed with CR and MO were desorbed using 0.1 mol L^−1^ sodium hydroxide (NaOH), and then the resultant NFs were washed with ultrapure water until neutral pH and loaded with GO/TiO_2_. The prepared composite NFs were then used in the photodegradation of organic dye solutions to explore the recycling performance of the composite NF membrane.

Calculation of adsorption capacity: The absorbance values of CR (497 nm) and MO (463 nm) solutions at different concentrations (0–400 mg/L) were measured by a UV-Vis spectrophotometer, and a standard curve was drawn. The absorbance of CR and MO solution before and after adsorption was measured and substituted into the standard curve to calculate the concentration of CR and MO solution before and after adsorption, and then the adsorption amount of CR and MO was calculated.

## 3. Results and Discussion

### 3.1. Structural Characterization of PLLA-CS Nanofibrous

Figure 1a,b shows the SEM images of core-shell structured PLLA-CS nanofibers prepared using our optimized electrospinning conditions. Figure 1d shows the diameter distribution of coaxial electrospun fibers. There are no polymer beads in the fiber. The fiber diameter distribution is relatively uniform, with an average diameter of 180.13 nm.

In order to observe the core/shell structure of the nanofibers, we carried out TEM analysis on the fiber samples. In Figure 1c, it can be seen that the fiber is composed of a darker core area and a brighter shell area, and the boundary between the core and the shell is clearly observed. This shows that PLLA nanofibers are completely covered by CS.

### 3.2. Structural Characterization of GO/TiO_2_

Figure 2a,b show SEM images of GO/TiO_2_ prepared from Ti(OBu)_4_ (1 mL), GO (8 mg), and HF (0.3 mL) as a morphology control agent. As can be seen from the figure, when 0.3 mL of HF was used, more {001} crystalline composites (typical spindle morphology) were obtained. As seen in Figure 2c, TiO_2_ grows on graphene with particle deposition. Figure 2d shows a TEM image of the GO/TiO_2_ prepared using 0.3 mL of HF, with the material exhibiting clear striped lattices with a spacing of 0.37 nm, corresponding to the {001} crystal plane of anatase TiO_2_, which indicates that TiO_2_ with exposed (110) plane was successfully synthesized, and the strong bonding together of GO and TiO_2_.

Figure 3 show XRD patterns of GO/TiO_2_ prepared by mixing Ti(OBu)_4_ (1 mg), GO (8 mg), and HF (0.3 mL). It can be seen from the patterns that the three prepared materials have anatase-type structures. The weak diffraction peak of GO at 2θ = 23.0° cannot be observed in the patterns, as it is superimposed by the strong diffraction peak of TiO_2_ at 25.3°. When 0.3 mL of HF was used in the preparation of HF, the peak at 25.3° is sharp and intense, which indicates that the crystallinity of TiO_2_ is good.

### 3.3. Structural Characterization PLLA/CS-GO/TiO_2_

Figure 4a shows the SEM image of GO/TiO_2_ on a PLLA/CS NF after 10 min of loading and Figure 4b shows an SEM image of GO/TiO_2_ on a PLLA/CS NF after 30 min of loading. As can be seen from the images, after a loading time of 10 min, only a small amount of GO/TiO_2_ is loaded on PLLA/CS, with relatively uniform distribution. Upon an increase in the loading time to 30 min, the amino groups on the NFs bond with GO/TiO_2_, with a relatively dense and uniform distribution. In addition, it can be seen from Figure 4 that GO and PLLA/CS nanofibers are tightly bound, which is mainly because the -NH_2_ on CS and a large number of -COOH groups on GO undergo amidation to form -NHCO-, which makes CO tightly bound to the surface of PLLA/CS nanofibers.

Figure 5a shows the survey XPS spectra of GO, GO/TiO_2_ and PLLA/CS–GO/TiO_2_. The spectra show that GO contains only two elements, C (284.6 eV) and O (530.7 eV), and both GO/TiO_2_ and PLLA/CS–GO/TiO_2_ contain C (284.6 eV), O (530.7 eV), and Ti (458.7 eV). Ti is more scattered on the PLLA/CS fiber membrane, so the intensity of the peak in the PLLA/CS–GO/TiO_2_ spectrum is weaker. In order to better reveal the changes of surface-active groups during the whole preparation process of the materials, the C 1s of GO, GO/TiO_2_, and PLLA/CS-GO/TiO_2_ were analyzed by XPS high-resolution spectroscopy. Figure 5b shows the high-resolution C 1s XPS spectrum of GO with hybridized C–C peaks at 284.8 eV, C–O–C or C–OH at 286.8 eV, and oxygen-containing groups such as O=C–OH at 288.8 eV. Figure 5c shows the high-resolution C 1s XPS spectrum of GO/TiO_2_, that the symmetric sp^2^ hybridized C-C peak still exists, and that the peak intensities of other reactive groups such as epoxy groups drop sharply. It is shown that most of GO is reduced during the preparation of GO-TiO_2_, resulting in a gradual reduction of oxygen-containing active groups on the graphene surface. The unreduced GO active groups will undergo an amidation reaction with the amino group of CS during the subsequent composite process of GO/TiO_2_ and PLLA/CS fibers. At the same time, a faint peak appeared at 282.3 eV in the C 1s XPS spectrum of the GO-TiO_2_ sample, which can be attributed to the C-Ti bond. Graphene is a carbon material, during the solvothermal process, a small amount of C enters the lattice of TiO_2_ to bond with Ti. Figure 5d shows the high-resolution C 1s XPS spectrum of PLLA/CS–GO/TiO_2_, which features a C–C peak at 284.8 eV, C–O–C and C–OH peaks of CS and GO at 286.8 eV, and a C=O active group at 288.8 eV, the peak of which is more intense than that of GO/TiO_2_ due to the carbonyl group of PLLA.

Figure 6 shows the FTIR spectra of GO, GO/TiO_2_ and PLLA/CS–GO/TiO_2_. The peaks at 1055.75, 1222.58, 1726.00 and 3410.04 cm^−1^ can be attributed to the oxygen-containing functional groups of GO. It is obvious from the FTIR spectra that the oxygen-containing functional group absorption peaks of GO in the spectrum of GO/TiO_2_ are sharply decreased in intensity compared with those of GO, which proves that most of the epoxy and hydroxyl groups were reduced. Peaks that can be attributed to –OH can be observed in the spectrum of PLLA/CS–GO/TiO_2_ due to the hydroxyl groups present in the CS in the NF shell layer. A peak present in the same spectrum at 485.06 cm^−1^ is related to the vibrational absorption of O–Ti. A skeletal stretching vibration of unoxidized graphite can be observed at 1625.17 cm^−1^. The peak at 3400 cm^−1^ is mainly caused by the stretching vibration of −OH, −NH and −NH_2_. The GO surface contains a large amount of −OH, so there is a strong peak at 3400 cm^−1^. During TiO_2_ deposition on GO surface, −OH on the GO surface reacts with tetrabutyl titanate, so the 3400 cm^−1^ peak in GO/TiO_2_ disappears. In PLLA/CS-GO/TiO_2_, the presence of −OH, −NH and −NH_2_ in CS led to the peak at 3400 cm^−1^.

### 3.4. Adsorption Performance Study

Figure 7 and Figure 8 show the UV spectral absorption standard curves of MO and CR, respectively. It can be seen from the figure that the absorbance is linear with the concentration. Figure 9a shows the change in the amount of MO adsorbed by PLLA/CS–GO at different temperatures in line with an extension in the adsorption time. From this data, it can be seen that the best temperature for adsorption is 30 °C, at which the highest amount adsorbed was 79.7 mg g−1, and that the temperature and amount adsorbed were inversely proportional, and adsorption saturation was reached after 120 min. Figure 9b shows the change in the amount of CR adsorbed by the PLLA/CS–GO composite at different temperatures in line with an extension of the adsorption time. It is worth noting that the adsorption of MO and CR at ≥40 °C is poor, which can be attributed to the solubility of the organic dye solutions in the aqueous environment under high temperature conditions, making it difficult for dye molecules to adsorb to CS and GO. The temperature of the subsequent adsorption experiments in this paper is 30 °C.

Figure 10 shows the relationship between the adsorption amount of (a) MO and (b) CR with time. Figure 10a shows the adsorption of MO by PLLA/CS, PLLA/CS–GO and PLLA/CS–GO/TiO_2_ at pH = 7 and room temperature, which reached saturation at 170 min, 120 min and 250 min, with saturation adsorption amounts of 60.09, 78.25 mg g^−1^ and 153.22 mg g^−1^, respectively. Figure 10b shows the photodegradation of MO by PLLA/CS, PLLA/CS–GO and PLLA/CS–GO/TiO_2_. In terms of the adsorption of CR by the composites, the adsorption of CR by PLLA/CS and PLLA/CS–GO at pH = 7 and room temperature tended to saturate at around 240 and 180 min, respectively, with saturated adsorption amounts of 138.01 and 150.22 mg g^−1^, respectively. The removal amount of MO by PLLA/CS–GO/TiO_2_ approached saturation at 180 min with a saturated removal amount of 795.44 mg g^−1^; a value 657.43 mg g^−1^ higher than that of PLLA/CS. The removal amount of CR by PLLA/CS–GO/TiO_2_ is much better than that of MO, with the fundamental reason for this being that the photocatalytic oxidation of CR by TiO_2_ is more complete than that of MO, and the removal rate of CR by fibers is higher than that of MO.

Figure 11 shows the comparison of color changes after photodegradation of PLLA/CS-GO/TiO_2_ materials with different concentrations of MO (0.08 mg/mL–0.4 mg/mL) and CR (0.8 mg/mL–2 mg/mL) dye solutions for 300 min. It can be clearly seen from the figure that after catalytic adsorption, the color of MO and CR dyeing solution is obviously lighter, and the color of lower concentration CR dyeing solution becomes basically transparent at last. This also shows that the adsorption effect of the material on CR is better than that on MO.

Figure 12 shows a comparison of the adsorption rates of CR and MO during the four cycles of the material, from which it can be seen that PLLA/CS–GO/TiO_2_ is better at adsorbing CR than MO in the process of recycling, as the removal rate of CR over the first three cycles is >97%, whereas the removal rate of MO over the first three cycles is >90%. In the fourth cycle, the pollutant removal capacity of fiber composite began to decline significantly, but the removal rate remained above 80%. The reason why the removal efficiency gradually decreases in the process of material recycling may be due to a small amount of fiber degradation and the loss of GO/TiO2 during desorption and ultrapure water treatment.

## 4. Conclusions

NFs were prepared via a coaxial electrostatic spinning method using PLLA as the nucleus layer support and CS as the shell layer adsorption phase, with GO loaded on the NF surface to study the performance of the composite NFs towards the adsorption of organic dyes. Ti(OBu)_4_ and GO were used as raw materials and HF was used as a morphology control agent to prepare an GO/TiO_2_ composite with a {001} crystalline surface, which was loaded onto PLLA/CS NFs. The NFs were prepared using PLLA as it is a degradable material, which is non-polluting to the environment and strengthens the NFs when used as a fiber core layer. The CS in the shell layer of the NFs has good inherent adsorption properties, increasing the surface contact between CS and the pollutants through spinning, and is convenient for recycling. Loading GO with PLLA/CS ensures that the GO does not agglomerate and shows better adsorption performance towards organic dyes. The TiO_2_ loaded on the {001} surface endows the composite material with good photocatalytic activity; on the one hand, GO promotes the photocatalytic activity of TiO_2_, while on the other hand, the rich functional groups on GO bond with CS. GO/TiO_2_ is uniformly distributed on PLLA/CS without agglomeration, ensuring that the prepared composite does not end up as a secondary pollutant in the environment. PLLA/CS–GO/TiO_2_ can be recycled, and exhibits photocatalysis and very high adsorption performance towards the studied organic dyes, which indicates that it has great application prospects for use in the catalytic adsorption of organic dyes.

## Figures and Tables

**Figure 1 polymers-14-04622-f001:**
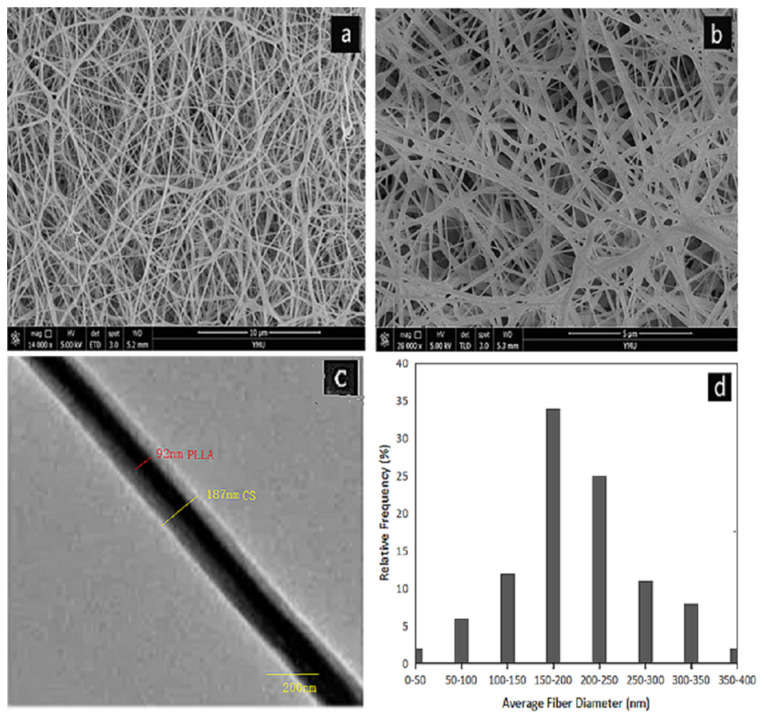
(**a**,**b**) SEM images of PLLA@CS coaxial nanofibers; (**c**) TEM images of PLLA@CS coaxial nanofibers; and (**d**) diameter distribution of PLLA@CS coaxial nanofibers;.

**Figure 2 polymers-14-04622-f002:**
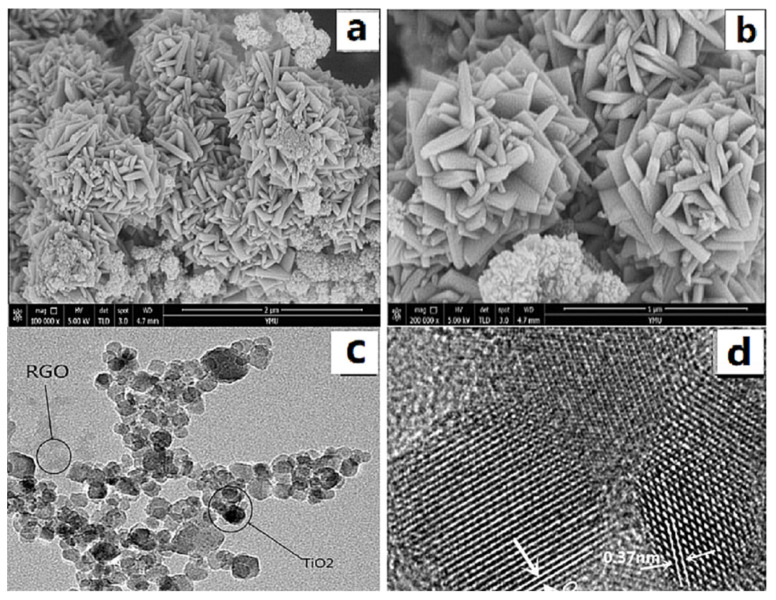
(**a**,**b**) SEM and (**c**,**d**) TEM images of GO/TiO_2_.

**Figure 3 polymers-14-04622-f003:**
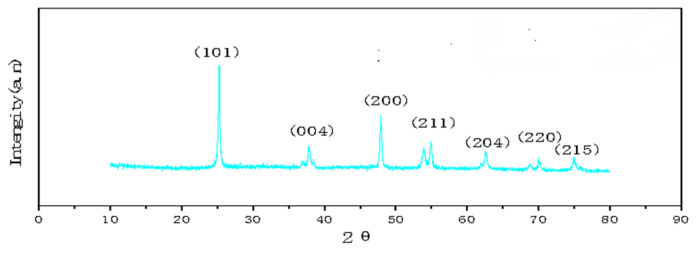
XRD patterns of GO/TiO_2_ samples.

**Figure 4 polymers-14-04622-f004:**
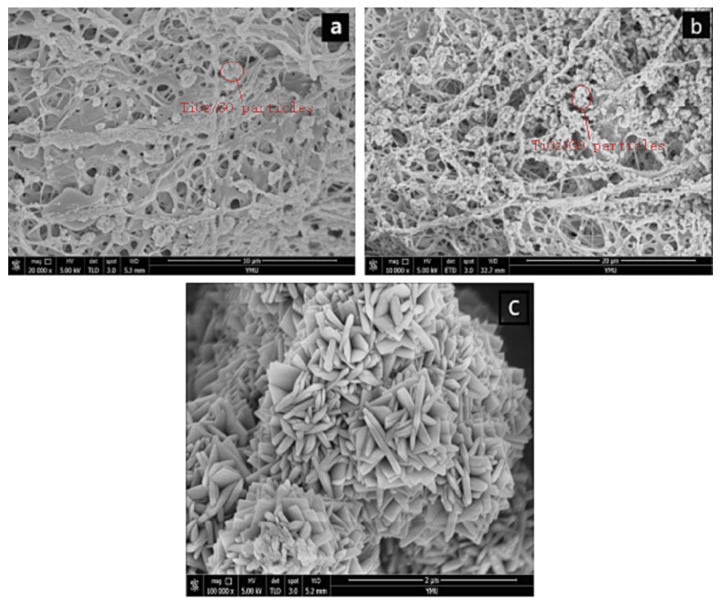
SEM images of PLLA/CS–GO/TiO_2_ after (**a**) 10 min and (**b**) 30 min of loading; (**c**) SEM image of TiO_2_ particles on fiber surface.

**Figure 5 polymers-14-04622-f005:**
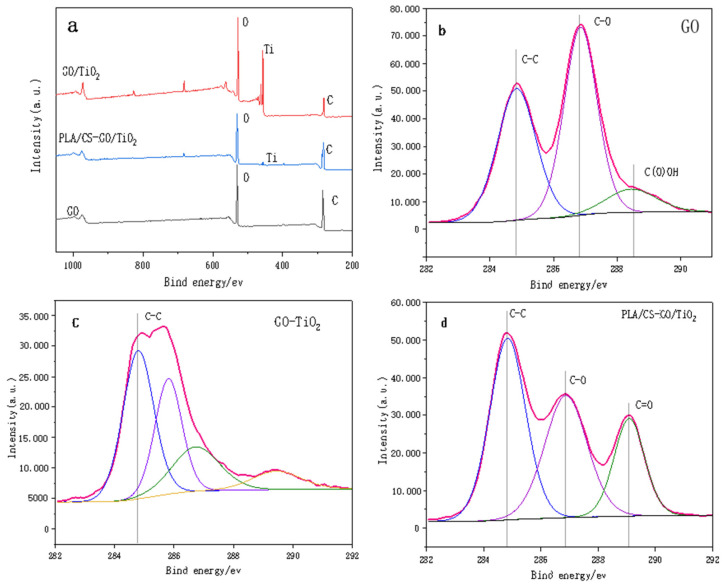
(**a**) Survey XPS spectra of GO, GO and PLLA/CS–GO/TiO_2_. High-resolution C 1s XPS spectra of (**b**) GO, (**c**) GO/TiO_2_ and (**d**) PLLA/CS–GO/TiO_2_.

**Figure 6 polymers-14-04622-f006:**
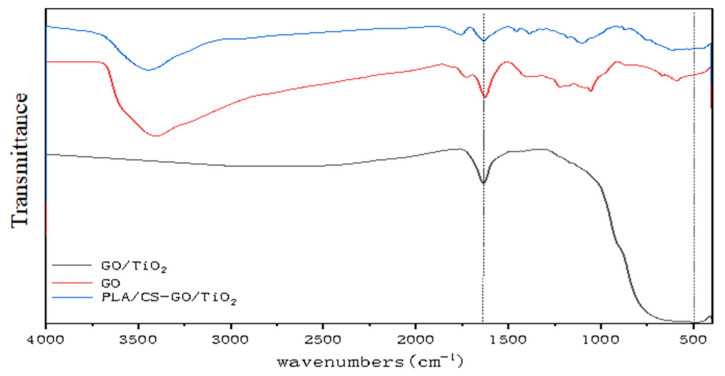
FTIR spectra of GO, GO/TiO_2_, and PLLA/CS–GO/TiO_2_.

**Figure 7 polymers-14-04622-f007:**
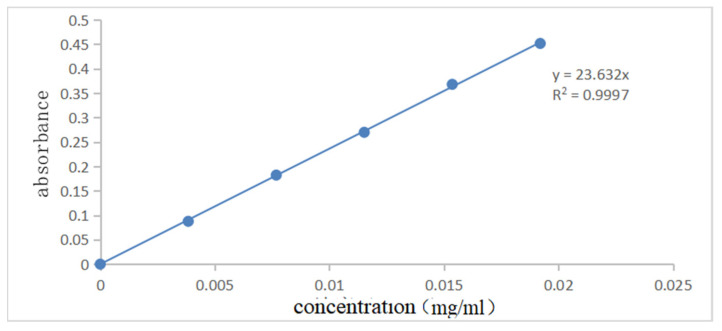
Standard curve of UV spectral absorption of MO (460 nm).

**Figure 8 polymers-14-04622-f008:**
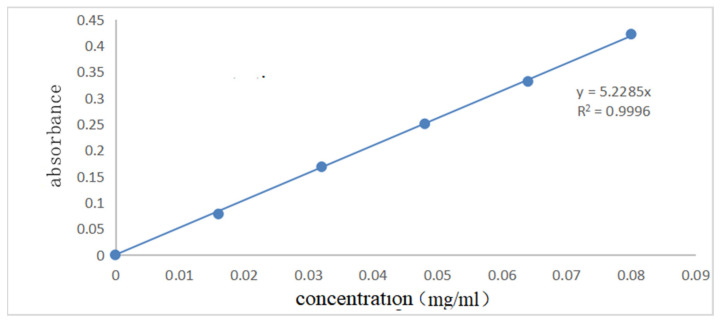
Standard curve of ultraviolet spectral absorption of CR (340 nm).

**Figure 9 polymers-14-04622-f009:**
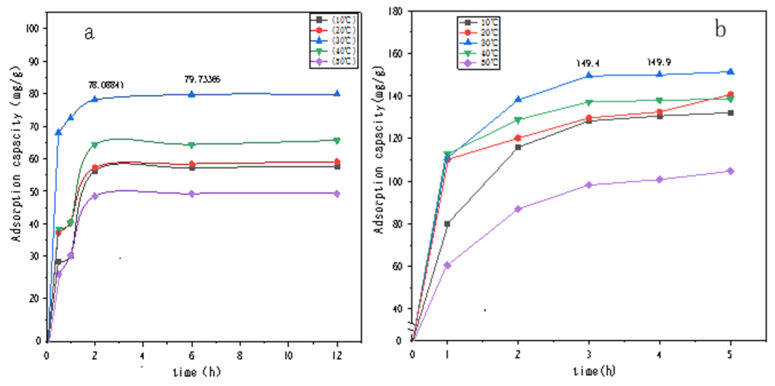
Effect that temperature and adsorption time have on the adsorption of (**a**) MO and (**b**) CR dye solutions by PLLA/CS–GO. mg/g refers to the mass (mg) of pollutants removed by the fiber composite material per unit mass (g).

**Figure 10 polymers-14-04622-f010:**
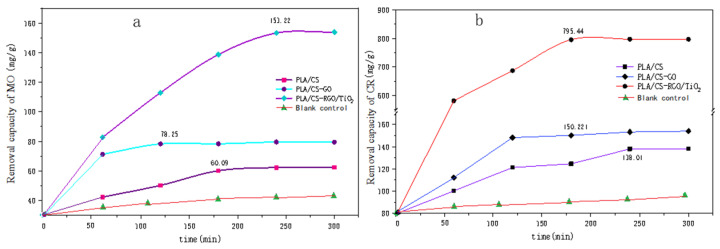
Removal capacity of (**a**) MO and (**b**) CR by PLLA/CS, PLLA/CS–GO, and PLLA/CS–GO/TiO_2_. mg/g refers to the mass (mg) of pollutants removed by the fiber composite material per unit mass (g).

**Figure 11 polymers-14-04622-f011:**
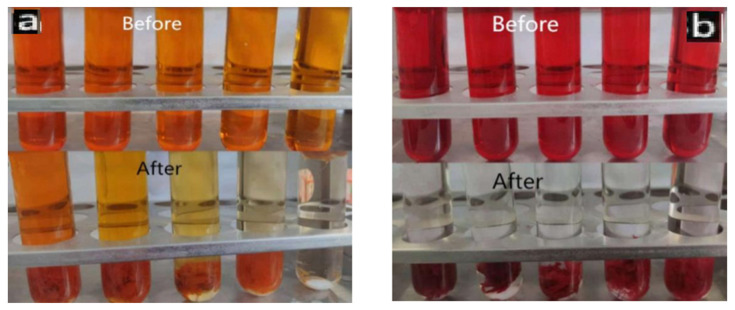
Color changes of MO (**a**) and CR (**b**) dye solutions before and after fiber photocatalysis.

**Figure 12 polymers-14-04622-f012:**
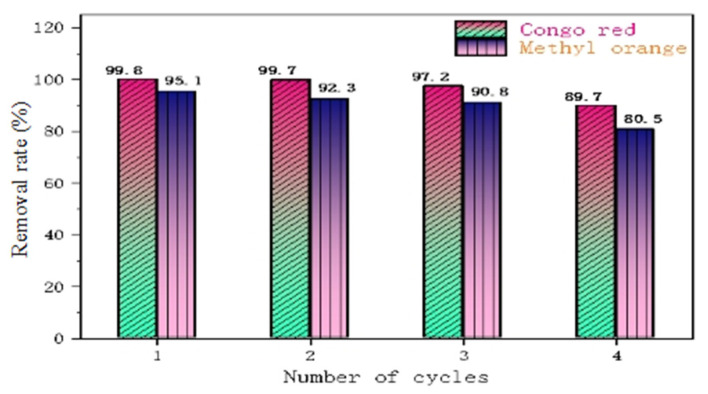
Removal performance of PLLA/CS–GO/TiO_2_ towards MO and CR over four cycles.

## Data Availability

The data presented in this study are available on request from the corresponding author.

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
