# Peer review of "Removal of Organic Pollutants with Polylactic Acid-Based Nanofiber Composites"

_polymers, 2022, doi:10.3390/polym14214622_

Round 1
Reviewer 1 Report
The title is "Removal of organic pollutants with polylactic acid-based nanofiber composites"
I do not recommend this manuscript for publication. TiO2 and its composites with graphene, RGO, and others are well known.
There are several issues with this manuscript:
What was the reason of use coaxial structure? The core-shell structure of nanofibers was not proven.
Chitosan is soluble in water, especially chitosan nanofibers. Were the nanofibers stable in water?
Why did the authors call photoadsorption? Upon irradiation of TiO2 with UV a photodegradation occurred.
line 211 "The fiber is destroyed by an external force, and the SEM image shown in Figure 1(d)"
what kind of force? How can you prove what is chitosan and PLA by SEM?
what is the wavelength of UV used in the experiments?
why not presented the control experiments of light irradiation of dyes? UV light irradiation will lead to photodegradation of dye. UV-Vis spectra of dyes are also not presented.
Figure 1. The caption is not clear.
the histogram of size distribution is not provided.
line 206 "hollows with an average 206 diameter of about 152 nm" where are hollows of 152 nm? on Figure 1.b I see 10 micrometers scale
figure 5. inscriptions are unreadable
figure 7 time intervals started not from 0.
Author Response
October 1, 2022
Dear Sir/Madam,
Thank you for reviewing our manuscript and offering valuable advice. In accordance with your suggestions, we have made the following revisions to our manuscript:
I do not recommend this manuscript for publication. TiO2 and its composites with graphene, RGO, and others are well known.
Response:As you said, TiO2 and its composites with graphene, RGO, etc. are currently widely used photocatalytic materials. We apologize for not expressing our work clearly.
Use of nanomaterials such as TiO2 in organic pollutants treatment, however, has its downfalls, reducing their potential in real-world applications. Not only are nanomaterials difficult, energy-intensive, and costly to retrieve from water post-treatment, their physicochemical prop-
erties can potentially induce aggregation during use. In addition, there are possible limitations to free nanoparticles’accepted use in water treatment devices due to conceivable exposures, and subsequently, potential toxicological risks. A promising method to mitigate these limitations is to restrain nanomaterial mobility by immobilizing nanomaterials on larger substrates or scaffolds. Anchoring nanomaterial to a substrate not only limits nanomaterial aggregation, but also facilitates the facile removal and reuse of the nanomaterial posttreatment.
The significance of our work is to prepare a composite material with biodegradable PLA/CS electrospun scaffold as the nano catalyst carrier to solve the above problems.
We have revised the introduction part of the manuscript to make the expression clearer and more accurate. Thank you very much for your support.
There are several issues with this manuscript:
What was the reason of use coaxial structure? The core-shell structure of nanofibers was not proven.
Response:That is a very good question. We apologize for not expressing ourselves clearly.
(1)The purpose of coaxial spinning is to ensure that chitosan can uniformly exist on the surface of the fiber scaffold to facilitate the adsorption of GO.
(2)We mainly used two methods to prove the successful preparation of the core-shell structure. First of all, after the fibers were dissolved with dichloromethane, obvious voids in the core layer could be observed on the SEM images; Secondly, after the fiber is broken by external force, it is obvious that the shell is torn and separated from the core layer.
Chitosan is soluble in water, especially chitosan nanofibers. Were the nanofibers stable in water?
Response:In our work, chitosan with 90% deacetylation degree is selected, which is almost insoluble in water.
Why did the authors call photoadsorption? Upon irradiation of TiO2 with UV a photodegradation occurred.
Response: Thank you very much for finding this error. We have changed the photoasorption in the article to photodegradation.
line 211 "The fiber is destroyed by an external force, and the SEM image shown in Figure 1(d)"
what kind of force? How can you prove what is chitosan and PLA by SEM?
Response:Thank you for your insightful comment.
- We use the universal tensile testing machine to break the fiber membrane, and the peeling of the shell and core layer can be observed through SEM at the fracture surface of the fiber. We believe that this kind of peeling phenomenon will not occur on the fracture surface of uniform single-layer fibers, which can prove that we have successfully prepared double-layer fibers.
- In the coaxial spinning process, we use PLLA as the core solution and CS as the shell solution. Therefore, the outer layer in Figure 1D is chitosan and the inner layer is PLLA.
what is the wavelength of UV used in the experiments?
Response:Thank you very much for your valuable comments, the UV wavelength used in the experiment is 350nm. We have added it to the manuscript.
why not presented the control experiments of light irradiation of dyes? UV light irradiation will lead to photodegradation of dye. UV-Vis spectra of dyes are also not presented.
Response:Thank you very much for your valuable comments.
- According to your suggestion, we have supplemented the control test of dye photodegradation and added it to the manuscript.
- According to your suggestion, we have supplemented the UV absorption spectrum data of the dye in the manuscript.
Figure 1. The caption is not clear.
Response:Thank you very much for your valuable comments. According to your suggestion, we rewrote the caption of Figure 1 in the manuscript.
the histogram of size distribution is not provided.
Response:Thank you very much for your valuable comments. According to your suggestion, we have supplemented the fiber size distribution histogram in the manuscript.
line 206 "hollows with an average 206 diameter of about 152 nm" where are hollows of 152 nm? on Figure 1.b I see 10 micrometers scale
Response:Thank you very much for finding this error. Please forgive us for our carelessness. We put the wrong SEM picture here. I uploaded the correct picture again, and marked the hole size on the picture.
figure 5. inscriptions are unreadable
Response: Thank you very much for your valuable comments.According to your suggestion, we have reprocessed Figure 5.
figure 7 time intervals started not from 0
Response:Thank you very much for your valuable comments.According to your suggestion, we have reprocessed Figure 7.
Thank you again for your valuable comments and suggestions.
Yours sincerely,
Dengbang Jiang
Reviewer 2 Report
This paper is on a bio-inspired nanofiber loaded with oxides for dye decomposition. The experiments are good and deliver new information. The following minor corrections are suggested.
1. In the Introduction part, some writing contains mistakes. For example, on the citation of [10], the authors say: Gan et al., but [10] is not from Gan's work.
2. "et al" may be corrected as "et al.".
3. There is a mistake on line 70. What is the meaning of "...fibers. choose. For example, the primary amine"?
4. Is there any difference between "PLA" and "PLLA"?
5. On line 125, "stably dispersed In the water." should be "stably dispersed in the water."
6. In Experimental section, the frequency unit "KHz" should be corrected as "kHz".
7. The unit for power should be "W" instead of "w" on line 148.
Author Response
October 1, 2022
Dear Sir/Madam,
Thank you for reviewing our manuscript and offering valuable advice. In accordance with your suggestions, we have made the following revisions to our manuscript:
- In the Introduction part, some writing contains mistakes. For example, on the citation of [10], the authors say: Gan et al., but [10] is not from Gan's work.
Response: Thank you very much for finding this error. I examined the manuscript carefully and corrected these mistakes.
- "et al" may be corrected as "et al.".
Response: Thank you very much for finding this error. I examined the manuscript carefully and corrected these mistakes.
- There is a mistake on line 70. What is the meaning of "...fibers. choose. For example, the primary amine"?
Response: Thank you very much for finding this error. I examined the manuscript carefully and corrected this mistake.
- Is there any difference between "PLA" and "PLLA"?
Response: Thank you very much for finding this error. All PLA in the manuscript refers to PLLA (Except for the introduction part). I carefully checked the manuscript and changed all PLA in the manuscript to PLLA.
- On line 125, "stably dispersed In the water." should be "stably dispersed in the water."
Response: Thank you very much for finding this error. I examined the manuscript carefully and corrected this mistake.
- In Experimental section, the frequency unit "KHz" should be corrected as "kHz".
Response: Thank you very much for finding this error. I examined the manuscript carefully and corrected these mistakes.
- The unit for power should be "W" instead of "w" on line 148.
Response: Thank you very much for finding this error. I examined the manuscript carefully and corrected this mistake.
Thank you again for your valuable comments and suggestions.
Yours sincerely,
Dengbang Jiang
Reviewer 3 Report
Obs.1
The state-of-the-art chapter must be improved and additional info inserted. The provided information from the ” Introduction” section looks more likely a technical report rather than a scientific description. Also, the authors must better address (in terms of the state-of-the art which is not up to date) the use of functionalized nanocomposites for the removal of organic pollutants (as a quick example, please see: https://doi.org/10.1007/978-3-030-76008-3_6).
Obs.2.
In terms of organic dyes (for example Rhodamine, as the authors mentioned in the “Introduction” section), it was demonstrated that a catalyzed oxidation reaction (using catalysts having rice husk at raw material) was more efficient that the adsorption on sugar-based AC, as authors mentioned (please see:
https://doi.org/10.3390/catal11070815). I suggest to the authors to be critical in terms of the obtained values in comparison with other groups results.
Obs.3
- Poor resolution for fig.5 and fig.7.
- “wavenumbers” is spelled in 1 word (fig.6)
Obs.4.
Do you have an explanation for the disappearances of peaks and bands from fig.6? for example the band from 3400-3500 cm-1.
Author Response
October 1, 2022
Dear Sir/Madam,
Thank you for reviewing our manuscript and offering valuable advice. In accordance with your suggestions, we have made the following revisions to our manuscript:
Obs.1
The state-of-the-art chapter must be improved and additional info inserted. The provided information from the ” Introduction” section looks more likely a technical report rather than a scientific description. Also, the authors must better address (in terms of the state-of-the art which is not up to date) the use of functionalized nanocomposites for the removal of organic pollutants (as a quick example, please see:https://doi.org/10.1007/978-3-030-76008-3_6).
Response: Thank you very much for your valuable comments.We carefully reviewed the relevant literature on the use of functional nanocomposites to remove organic pollutants, and cited the review written by Niculescu et al [7].
Obs.2.
In terms of organic dyes (for example Rhodamine, as the authors mentioned in the “Introduction” section), it was demonstrated that a catalyzed oxidation reaction (using catalysts having rice husk at raw material) was more efficient that the adsorption on sugar-based AC, as authors mentioned (please see:https://doi.org/10.3390/catal11070815). I suggest to the authors to be critical in terms of the obtained values in comparison with other groups results.
Response: Thank you very much for your valuable comments.We carefully reviewed the relevant literature you recommended and cited the work done by Niculescu et al. [21].
Obs.3
Poor resolution for fig.5 and fig.7.
Response: Thank you very much for your valuable comments.According to your suggestion, we have reprocessed fig.5 and fig.7.
“wavenumbers” is spelled in 1 word (fig.6)
Response: Thank you very much for finding this error. I examined the manuscript carefully and corrected this mistake.
Obs.4.
Do you have an explanation for the disappearances of peaks and bands from fig.6? for example the band from 3400-3500 cm-1.
Response:That is a very good question. We apologize for not expressing ourselves clearly.
The peak at 3400cm-1 is mainly caused by the stretching vibration of - OH, - NH and - NH2. The GO surface contains a large amount of - OH, so there is a strong peak at 3400cm-1. During TiO2 deposition on GO surface, - OH on GO surface reacts with tetrabutyl titanate, so the 3400cm-1 peak in GO/TiO2 disappears. In PLLA/CS-GO/TiO2, the presence of -OH,- NH and - NH2 in CS led to the peak at 3400 cm-1.
We have revised the corresponding part in the manuscript to make the expression clearer and more accurate.
Thank you again for your valuable comments and suggestions.
Yours sincerely,
Dengbang Jiang
Round 2
Reviewer 1 Report
Here is an example of immobilized TiO2/rGO particles with high photocatalytic activity:
https://doi.org/10.3390/catal9090708, no issues with aggregation.
A)
"Response: In our work, chitosan with 90% deacetylation degree is selected, which is almost insoluble in water."
Chitosan with 90% deacetylation is very well soluble in water.
A higher degree of deacetylation = better solubility. For example https://pubs.acs.org/doi/10.1021/bm000036j
B)
Figure 1:
"Cross section of fiber" Do you mean a single fiber? in this case, the scale bar on Fig.1(b) is 2 µm. and the authors state that the diameter of the fiber is 152 nm.
I understand that after DCM treatment you will lose the fiber structure, but this is not a provement of core-shell structure. If you have a core-shell structure - what is the diameter of the core and the shell?
How this SEM image Fig.1(b) can prove the core-shell structure?
Figure 1 caption "(c) 100000×SEM image" of what SEM image? Dissolved /pristine?
Fig.1(c) is not even discussed in the text
Fig.1(d) is not a cross-section also. If you want to prove that core-shell TEM is required of a FIB-SEM.
Fig1.d looks like this is a flat belt structure of the fiber but not the core-shell. Anyway from SEM, it is not possible to prove the material.
Line 211:
"As shown in Fig.1(b), hollows with an average 211
Diameter of about 152 nm were produced on the nanofibers, and the hollow part of the
Fibers was generated by the dissolution of PLLA, which proved that the prepared PLLA-
CS nanofibers were the shell-core structure of CS-coated PLLA"
"PLLA (0.3 g) was dissolved in HFIP (4 mL) to prepare a nucleus layer solution with 115
a concentration of 0.075 g mL−1 Separately, CS (0.15 g) was dissolved in a mixed solution 116
of HFIP (6 mL) and FA(1 mL) "
PLLA is soluble in HFIP, usually, if you want to prepare core-shell nanofibers you need non-miscible liquids.
the authors have HFIP in both liquids, so PLLA will dissolve in both liquids. I can accept that you have blended electrospinning but not core-shell.
C) "Figure 7. FTIR spectra of GO, GO/TiO2, and PLLA/CS–GO/TiO2."
This is transmittance, not absorbance.
D) the resolution of fig 5 is too low.
what are particles and what is fibers in fig 5?
why after 30 min loading the particles have a sparser structure than after 10 min?
E) "Response:Thank you very much for your valuable comments. According to your suggestion, we have reprocessed Figure 7."
The figure was not corrected, again you have started not from 0.
F) Figure 11. photodegradation of (a) MO and (b) CR by PLLA/CS, PLLA/CS–GO, and 341
PLLA/CS–RGO/TiO2
Figure 11 is called "photodegradation " and the y-axis is "adsorption capacity"? so how do you measure adsorption or photodegradation, I guess these are different processes.
what does the y-axis mean in the case of control? like mg/g of what?
G) some interesting works about photodegradation and electrospinning/blend electrospinning can be useful for the readers:
1) Zouzelka, R.; Remzova, M.; Plsek, J.; Brabec, L.; Rathousky, J. Catalysts 2019, 9 (9), 708.
2) Elashnikov, R.; Rimpelová, S.; Vosmanská, V.; Kolská, Z.; Kolářová, K.; Lyutakov, O.; Švorčík, V. React. Funct. Polym. 2019, 143.
3) Kutorglo, E. M.; Elashnikov, R.; Rimpelova, S.; Ulbrich, P.; Říhová Ambrožová, J.; Svorcik, V.; Lyutakov, O. ACS Appl. Mater. Interfaces 2021, 13 (14), 16173–16181.
4) Elashnikov, R.; Rimpelová, S.; Lyutakov, O.; Pavlíčková, V. S.; Khrystonko, O.; Kolská, Z.; Švorčík, V. ACS Appl. Bio Mater. 2022, 5 (4), 1700–1709.
5) DOI: 10.1039/C7AN00419B. Analyst 2017, 142 (16), 2974–2981.
H) plenty of shortcomings and inconsistencies should be corrected:
u have both Figure 1 and Fig(1), 152 nm and 340nm / 0.8mg/ml-2mg/ml (with space without)
Fig(1) has different sizes of cases of letters and some lowercase some upper
(I) English must be improved, e.g:
"Diameter of about 152 nm were produced on the nanofibers, "
"Fibers was generated "
"The shell-core structure" is usually core-shell, but in your case blended is better
and other
Author Response
October 13, 2022
Dear Sir/Madam,
Thank you for reviewing our manuscript and offering valuable advice. In accordance with your suggestions, we have made the following revisions to our manuscript:
Here is an example of immobilized TiO2/rGO particles with high photocatalytic activity:
https://doi.org/10.3390/catal9090708, no issues with aggregation.
Response: I apologize for not expressing ourselves clearly. In the introduction section, we supplemented relevant references on agglomeration and difficulty in recycling of nano titanium dioxide.
https://doi.org/10.1016/j.apt.2013.05.017
https://doi.org/10.1016/j.watres.2010.09.013
https://pubs.acs.org/doi/10.1021/es062726m
A)"Response: In our work, chitosan with 90% deacetylation degree is selected, which is almost insoluble in water."
Chitosan with 90% deacetylation is very well soluble in water. A higher degree of deacetylation = better solubility. For example https://pubs.acs.org/doi/10.1021/bm000036j
Response:Your question is crucial to whether the fiber we prepared can be used for water treatment. We apologize for not expressing ourselves clearly. We have carefully reviewed a large amount of literature on the solubility of chitosan. Chitosan can only dissolve in some specific organic acids and a few inorganic solutions. The chitosan (Mw=100000Da, 90%deacetylated) we use is almost insoluble in water.
For example:
https://www.nature.com/articles/pj199721/
https://doi.org/10.1016/j.carres.2009.03.002
https://doi.org/10.1016/j.progpolymsci.2006.06.001
B)
Figure 1: "Cross section of fiber" Do you mean a single fiber? in this case, the scale bar on Fig.1(b) is 2 µm. and the authors state that the diameter of the fiber is 152 nm. I understand that after DCM treatment you will lose the fiber structure, but this is not a provement of core-shell structure. If you have a core-shell structure - what is the diameter of the core and the shell?
How this SEM image Fig.1(b) can prove the core-shell structure?
Figure 1 caption "(c) 100000×SEM image" of what SEM image? Dissolved /pristine?
Fig.1(c) is not even discussed in the text
Fig.1(d) is not a cross-section also. If you want to prove that core-shell TEM is required of a FIB-SEM.
Fig1.d looks like this is a flat belt structure of the fiber but not the core-shell. Anyway from SEM, it is not possible to prove the material.
Response:Thank you for your insightful comment.
- According to your valuable suggestions, we took TEM images of fibers. The TEM diagram of fiber is added to Figure 1.
- We have deleted Figure 1 (b), Figure 1 (c) and Figure 1 (d) which cannot be used to characterize the fiber core shell structure in the manuscript.
(3)From TEM photos, we can see the core shell structure of the fiber directly.
Thank you again for your valuable suggestions.
Line 211:
"As shown in Fig.1(b), hollows with an average diameter of about 152 nm were produced on the nanofibers, and the hollow part of the fibers was generated by the dissolution of PLLA, which proved that the prepared PLLA CS nanofibers were the shell-core structure of CS-coated PLLA"
"PLLA (0.3 g) was dissolved in HFIP (4 mL) to prepare a nucleus layer solution with a concentration of 0.075 g mL−1 Separately, CS (0.15 g) was dissolved in a mixed solution of HFIP (6 mL) and FA(1 mL) "
PLLA is soluble in HFIP, usually, if you want to prepare core-shell nanofibers you need non-miscible liquids.
the authors have HFIP in both liquids, so PLLA will dissolve in both liquids. I can accept that you have blended electrospinning but not core-shell.
Response:This is a good question. The two soluble solutions can be used for coaxial spinning. For example, when DMF (or HFIP) was used as the internal and external solvent, coaxial fibers with good internal and external bonding can be formed.
https://doi.org/10.1016/j.ceramint.2019.12.058
https://doi.org/10.1016/j.msec.2013.09.020
https://doi.org/10.1016/j.compscitech.2016.02.005
- C) "Figure 7. FTIR spectra of GO, GO/TiO2, and PLLA/CS–GO/TiO2."
This is transmittance, not absorbance.
Response: Thank you very much for finding this error. I have corrected this mistake in the manuscript.
- D) the resolution of fig 5 is too low.
what are particles and what is fibers in fig 5?
why after 30 min loading the particles have a sparser structure than after 10 min?
Response: Thank you very much for your valuable comments. We reprocessed Figure 5 and labeled TiO2/GO nanoparticles on it.
- E) "Response:Thank you very much for your valuable comments. According to your suggestion, we have reprocessed Figure 7."
The figure was not corrected, again you have started not from 0.
Response:Thank you very much for your valuable comments. According to your suggestion, we have reprocessed the Figure.
- F) Figure 11. photodegradation of (a) MO and (b) CR by PLLA/CS, PLLA/CS–GO, and 341
PLLA/CS–RGO/TiO2
Figure 11 is called "photodegradation " and the y-axis is "adsorption capacity"? so how do you measure adsorption or photodegradation, I guess these are different processes.
what does the y-axis mean in the case of control? like mg/g of what?
Response:Thank you very much for your valuable comments. I'm sorry that the statement here in the manuscript is inaccurate. We have rewritten this part according to your questions.
Here, we do not distinguish between photodegradation and adsorption, but use the pollutant treatment capacity per unit mass of composite fiber materials to measure the pollutant removal performance of materials. The y-axis in the figure is the pollutant treatment capacity of composite fiber material per unit mass. mg/g refers to the mass (mg) of pollutants removed by the fiber composite material per unit mass (g).
- G) some interesting works about photodegradation and electrospinning/blend electrospinning can be useful for the readers:
1) Zouzelka, R.; Remzova, M.; Plsek, J.; Brabec, L.; Rathousky, J. Catalysts,2019, 9 (9), 708.
2) Elashnikov, R.; Rimpelová, S.; Vosmanská, V.; Kolská, Z.; Kolářová, K.; Lyutakov, O.; Švorčík, V. React. Funct. Polym. 2019, 143.
3) Kutorglo, E. M.; Elashnikov, R.; Rimpelova, S.; Ulbrich, P.; Říhová Ambrožová, J.; Svorcik, V.; Lyutakov, O. ACS Appl. Mater. Interfaces 2021, 13 (14), 16173–16181.
4) Elashnikov, R.; Rimpelová, S.; Lyutakov, O.; Pavlíčková, V. S.; Khrystonko, O.; Kolská, Z.; Švorčík, V. ACS Appl. Bio Mater. 2022, 5 (4), 1700–1709.
5) DOI: 10.1039/C7AN00419B. Analyst 2017, 142 (16), 2974–2981.
Response:Thank you very much for your valuable comments. I carefully read and quoted the paper you recommended. It will be of great help to the improvement of my manuscript and future research work.
- H) plenty of shortcomings and inconsistencies should be corrected:
u have both Figure 1 and Fig(1), 152 nm and 340nm / 0.8mg/ml-2mg/ml (with space without)
Fig(1) has different sizes of cases of letters and some lowercase some upper
(I) English must be improved, e.g:
"Diameter of about 152 nm were produced on the nanofibers, "
"Fibers was generated "
"The shell-core structure" is usually core-shell, but in your case blended is better
and other
Response:Thank you very much for finding this error. I checked the manuscript carefully and corrected the mistakes.
Yours sincerely,
Dengbang Jiang

Reviewer 3 Report
Page 4, line 177 – space between value and measurement unit
Page 8, line 294 – space between value and measurement unit
Reference 7 is the same as reference 38
Check ref. 39 (no reference is indicated in the Reference list); the manuscript must be revised in order to identify whether or not reference 39 exists; otherwise, the number 39 must be removed from the list.
Author Response
October 14, 2022
Dear Sir/Madam,
Thank you for reviewing our manuscript and offering valuable advice. In accordance with your suggestions, we have made the following revisions to our manuscript:
Page 4, line 177 – space between value and measurement unit
Page 8, line 294 – space between value and measurement unit
Reference 7 is the same as reference 38
Check ref. 39 (no reference is indicated in the Reference list); the manuscript must be revised in order to identify whether or not reference 39 exists; otherwise, the number 39 must be removed from the list.
Response:Thank you very much for finding this error. I checked the manuscript carefully and corrected the mistakes.
Yours sincerely,
Dengbang Jiang

Round 3
Reviewer 1 Report
Thanks to the authors for the improvements and for their work.
Reviewer 3 Report
I have no further comments.